# A Functional K^+^ Channel from Tetraselmis Virus 1, a Member of the *Mimiviridae*

**DOI:** 10.3390/v12101107

**Published:** 2020-09-29

**Authors:** Kerri Kukovetz, Brigitte Hertel, Christopher R. Schvarcz, Andrea Saponaro, Mirja Manthey, Ulrike Burk, Timo Greiner, Grieg F. Steward, James L. Van Etten, Anna Moroni, Gerhard Thiel, Oliver Rauh

**Affiliations:** 1Membrane Biophysics, Department of Biology, Technische Universität Darmstadt, 64287 Darmstadt, Germany; kukovetz@bio.tu-darmstadt.de (K.K.); hertelb@bio.tu-darmstadt.de (B.H.); manthey@bio.tu-darmstadt.de (M.M.); burk@bio.tu-darmstadt.de (U.B.); thiel@bio.tu-darmstadt.de (G.T.); 2Department of Oceanography, Daniel K. Inouye Center for Microbial Oceanography: Research and Education, University of Hawai’i at Mānoa, 1950 East-West Road, Honolulu, HI 96822, USA; schvarcz@gmail.com (C.R.S.); grieg@hawaii.edu (G.F.S.); 3Department of Biosciences and CNR IBF-Mi, Università degli Studi di Milano, Via Giovanni Celoria 26, 20133 Milano, Italy; andrea.saponaro@unimi.it (A.S.); anna.moroni@unimi.it (A.M.); 4University Clinic for Psychiatry and Psychotherapy, Brandenburg Medical School, Immanuel Klinik Rüdersdorf, Seebad 82/83, 15562 Rüdersdorf, Germany; timogreiner@googlemail.com; 5Department of Plant Pathology and Nebraska Center for Virology, University of Nebraska Lincoln, Lincoln, NE 68583-0900, USA; jvanetten1@unl.edu; 6Institute of Biophysics, Consiglio Nazionale delle Ricerche, Via Celoria 26, 20133 Milan, Italy

**Keywords:** viral K^+^ channel, Kcv, K^+^ channel evolution, *Phycodnaviridae*, *Mimiviridae*

## Abstract

Potassium ion (K^+^) channels have been observed in diverse viruses that infect eukaryotic marine and freshwater algae. However, experimental evidence for functional K^+^ channels among these alga-infecting viruses has thus far been restricted to members of the family *Phycodnaviridae,* which are large, double-stranded DNA viruses within the phylum *Nucleocytoviricota*. Recent sequencing projects revealed that alga-infecting members of *Mimiviridae*, another family within this phylum, may also contain genes encoding K^+^ channels. Here we examine the structural features and the functional properties of putative K^+^ channels from four cultivated members of *Mimiviridae*. While all four proteins contain variations of the conserved selectivity filter sequence of K^+^ channels, structural prediction algorithms suggest that only two of them have the required number and position of two transmembrane domains that are present in all K^+^ channels. After in vitro translation and reconstitution of the four proteins in planar lipid bilayers, we confirmed that one of them, a 79 amino acid protein from the virus Tetraselmis virus 1 (TetV-1), forms a functional ion channel with a distinct selectivity for K^+^ over Na^+^ and a sensitivity to Ba^2+^. Thus, virus-encoded K^+^ channels are not limited to *Phycodnaviridae* but also occur in the members of *Mimiviridae*. The large sequence diversity among the viral K^+^ channels implies multiple events of lateral gene transfer.

## 1. Introduction

After the discovery that the M2 protein of influenza virus A exhibits ion channel function [1], many more viruses were discovered to contain genes encoding channel-forming proteins [2,3,4,5,6]. Structure and function studies have shown that these proteins have little in common. In terms of function, the known viral channels exhibit a large diversity ranging from some with high ion selectivity [1,7] to a majority with non-selective channels [4,5]. Furthermore, the structures of viral channels are very diverse. The only common feature is that they all contain variable numbers of α-helical transmembrane domains and most are small, consisting of only about 100 amino acids [2,5].

The structures of many known viral channels have no resemblance to channel proteins from prokaryotes and eukaryotes [2,5,8]. A prominent exception is viral channels that have the structural hallmarks of prokaryotic and eukaryotic K^+^ channels. Genes encoding this type of protein have been detected in phages (*Vibrio* phages, *Mycobacterium* phages) and in giant dsDNA viruses infecting algae [8]. The first viral K^+^ channel gene was detected in the genome of the chlorovirus Paramecium bursaria Chlorella virus 1 (PBCV-1) [7]. Subsequent analyses confirmed that the viral gene product had the basic structural and functional hallmarks of the pore module of all K^+^ channels, including those in human neurons and muscles. The channel, named Kcv for K^+^ channel of chloroviruses, has two transmembrane domains that are separated by a stretch of amino acids containing the pore helix and the highly conserved selectivity filter of all K^+^ channels [9]. Four of these monomers assemble into a homotetramer, which creates a central water-filled pore for ion transport. Different from the related prokaryotic and eukaryotic Kir channels, which have the same pore module architecture, the PBCV-1 channel has no appreciable cytosolic domains [10], which accounts for its small size (93 amino acids).

A large number of studies have confirmed that Kcv proteins from many chloroviruses not only look like a K^+^ channel, but they also share the functional features of complex K^+^ channels from prokaryotes and eukaryotes [7,11,12]. This includes the same mechanism of ion selectivity, which provides the channel with a distinct preference for transport of K^+^ over Na^+^ as well as a sensitivity to Ba^2+^ block [7,9,12]. Moreover, like their prokaryotic and eukaryotic relatives, the viral channel is gated, meaning that it switches stochastically between defined open and closed states [12,13].

Intensive sequencing and analysis of environmental samples has led to the discovery of ca. 100 diverse K^+^ channel sequences in different species of *Phycodnaviridae* [6,8,14,15,16,17]. This includes not only viruses that infect unicellular *Chlorella*-like green algae in freshwater but also a range of viruses that have marine algae as hosts [16,17].

The widespread occurrence of viral K^+^ channels indicates that they are not just a peculiarity of one virus/host system. This also raises the question of the evolutionary origin of the viral K^+^ channels. The most obvious scenario would be that the viruses have obtained the K^+^ channel genes by molecular piracy from their host. However, a phylogenetic comparison of viral K^+^ channel sequences with those of two host cells revealed no evidence for such a recent gene transfer between host and virus [11,18]. Another open question in this context is whether genes encoding K^+^ channels are more broadly distributed in the phylum *Nucleocytoviricota* (also known as nucleocytoplasmic large DNA viruses (NCLDVs)) or unique to the family *Phycodnaviridae*. While the genomes of initially discovered amoeba-infecting mimiviruses contained no K^+^ channel sequences, recent data suggest that putative K^+^ channel genes are present in some alga-infecting members of the family *Mimiviridae*. These include possible K^+^ channel genes in Aureococcus anophagefferens virus (AaV), Tetraselmis virus 1 (TetV-1), and Organic Lake phycodnavirus 2 (OLPV2) viruses [8,19,20,21].

The annotation of genes in newly sequenced organisms relies on a few crucial hallmarks for a protein family, so sequence data alone does not guarantee that the gene product exhibits the annotated function. Fortunately, small channel proteins can easily be synthesized in vitro and functionally reconstituted in planar lipid bilayers [22]. This makes them ideal for testing whether the annotated function is real, which has important consequences for understanding their origins and mechanisms of action. Comparative analyses of the structure and function of K^+^-channel-like proteins from the different virus families, for example, will shed light on the evolution of these proteins within members of the *Mimiviridae* (mimivirids) and possibly even on the origin of all K^+^ channels. Furthermore, the minimal size, high structural variability, and robust function of viral K^+^ channels [6,12,14] makes these proteins valuable model systems for understanding basic the structure/function correlates of all K^+^ channel pores [12].

Here we tested the function of putative K^+^ channels from four alga-infecting mimivirids and confirmed channel function with a distinct selectivity for K^+^ over Na^+^ for one of them (TetV-1). The results of this study confirm that functional K^+^ channels are present not only in the members of *Phycodnaviridae*, but they also occur in viruses in the *Mimiviridae* family. The phylogenetic similarity between K^+^ channel genes from the different viruses advocates a common evolutionary origin.

## 2. Materials and Methods

### 2.1. Sequence Sources

The putative K^+^ channel sequences used in this study are from four mimivirids that infect marine phytoplankton (algae): TetV-1 [20], Florenciella virus Station ALOHA 1 (FloV-SA1), Rhizochromulina virus Station ALOHA 1 (RhiV-SA1), and Chrysochromulina sp. virus Kāne‘ohe Bay 1 (CsV-KB1) [23]. The virus-host systems derive from coastal waters of the island of O‘ahu (Kāne‘ohe Bay) in the State of Hawaii, USA (TetV-1, CsV-KB1) or an open ocean site (Station ALOHA) 100 km north of O‘ahu (FloV-SA1, RhiV-SA1). TetV-1 infects a strain of *Tetraselmis* [20]. CsV-KB1, FloV-SA1, and RhiV-SA1 infect unicellular algae in the genera *Chrysochromulina, Florenciella*, and *Rhizochromulina*, respectively [23]. Putative K^+^ channel genes encoded in the genomes of CsV-KB1, FloV-SA1, and RhiV-SA1 were predicted using Prodigal v2.6.3 [24] and identified based on protein sequence similarity searches employing webBLAST [25], CD-Search [26], Pfam [27], and InterProScan [28], as described in [23].

### 2.2. Sequence Analysis

Sequence alignment was performed with Clustal omega algorithm (https://www.ebi.ac.uk/Tools/msa/clustalo). Transmembrane domains were predicted by TMHMM (http://www.cbs.dtu.dk/services/TMHMM-2.0/), MINNOU (http://minnou.cchmc.org), TMpred: https://embnet.vital-it.ch/software/TMPRED_form, HMMTop: http://www.enzim.hu/hmmtop/ and SOSUI: http://harrier.nagahama-i-bio.ac.jp/sosui/sosui_submit.html algorithms. We used MEGA software version X to construct the phylogenetic trees [29]. Primary sequences of putative K^+^ channels were compared with sequences in databanks using the BLASTP algorithm (https://blast.ncbi.nlm.nih.gov/Blast.cgi?PROGRAM=blastp&PAGE_TYPE=BlastSearch&LINK_LOC=blasthome).

### 2.3. 3D Modeling

The 3D model structure for the TetV-1 protein was created by using the web portal for protein modeling Phyre2 [30] on the basis of the X-ray template structure of the K^+^ channel of *Streptomyces lividans* (KcsA; Protein Data Bank (PDB) ID: 1K4C). The 3D model was further refined [31] by using the web server 3Drefine [32].

### 2.4. Planar Lipid Bilayer Experiments

These experiments were done either on a vertical bilayer set up (IonoVation, Osnabrück, Germany) as described previously [6,22] or in horizontal bilayers using an eNPR amplifier, equipped with a BLM_chip flowcell (Elements srl, Cesena, Italy). A 1% hexadecane solution (MERCK KGaA, Darmstadt, Germany) in n-hexane (Carl ROTH, Karlsruhe, Germany) was used for pretreating the Teflon foil (Goodfellow GmbH, Hamburg, Germany). The hexadecane solution (ca. 0.5 μL) was added to the rim of the hole (100 μm in diameter) in the Teflon foil with a bent Hamilton syringe (Hamilton Company, Reno, NV, USA). The experimental solution contained 100 mM KCl and was buffered to pH 7.0 with 10 mM HEPES/KOH. As a lipid we used 1,2-diphythanoyl-*sn*-glycero-3-phosphocholine (DPhPC), 1,2-diphythanoyl-*sn*-glycerol-3 phosphatidyl-serine DPhPS, or n1,2-diphytanoyl-sn-glycero-3-phospho-(1′-rac-glycerol) DPhPG.

### 2.5. Channel Protein Synthesis

All channels were synthesized cell-free with the MembraneMax^TM^
*HN* Protein Expression Kit (Invitrogen, Carlsbad, CA, USA) as reported previously [22]. In vivo synthesis occurred on a shaker with 1000 rpm at 37 °C for 1.5 h in the presence of nanodiscs (NDs) with 1,2-dimyristoyl-*sn*-glycero-3-phosphocholine (DMPC) lipids (Cube Biotech GmbH, Monheim, Germany). The scaffold proteins of the NDs were His-tagged, which allowed purification of channel/ND complexes via metal chelate affinity chromatography. The concentration of His-tagged NDs in the reaction mixture was adjusted to 30 µM. For purification of the channel/ND complexes, the crude reaction mixture was adjusted to 400 µL with equilibration buffer (10 mM imidazole, 300 mM KCl, 20 mM NaH_2_PO_4_, pH 7.4 with KOH) and then loaded on an equilibrated 0.2 mL HisPur nickel–nitrilotriacetic acid agarose (Ni–NTA) spin column (Thermo Scientific). For binding of His-tagged NDs to the Ni–NTA resin, the columns were incubated for 45 min at RT and 200 rpm on an orbital shaker. In the subsequent step, the buffer was removed by centrifugation. To eliminate unspecific binders, the column was washed three times with 400 µL of a 20 mM imidazole solution. Finally, the His-tagged NDs were eluted in three fractions with 200 µL of a 250 mM imidazole solution. All centrifugation steps were performed at 700× *g* for 2 min. After purification, the preparations were stored at 4 °C. For the reconstitution of channel proteins into the lipid bilayer, a small amount (~2 µL) of the purified channel/ND conjugates (1:1000 dilutions) was added directly below the bilayer in the *trans* compartment.

### 2.6. Ion Channel Recordings, Data Analysis, and Statistics

After the insertion of an active channel into the bilayer, the membrane was routinely clamped for periods between 10 s and 15 min to a range of positive and negative voltages (usually from +160 mV to −160 mV in 20 mV steps). Data analysis was performed with KielPatch (version 3.20 ZBM/2011) and Patchmaster (HEKA Electronik, Lambrecht, Germany). Experimental data are presented as mean ± standard deviation (sd) of *n* independent experiments. Statistical significances were evaluated by one-way ANOVA and Student T-tests.

## 3. Results and Discussion

### 3.1. Putative K^+^ Channels

The amino acid sequences of the putative K^+^ channels from TetV-1, CsV-KB1, FloV-SA1, and RhiV-SA1 viruses exhibit a low degree of identity/homology with the exception of a small cluster of amino acids around the conserved motif (TXXTXGYG) characteristic of the selectivity filter of K^+^ channels (Figure 1). All of them have the conserved GYG sequence, but some have unusual substitutions or features at other positions. For example, the third variable position of the selectivity filter motif is often a threonine but may be substituted by serine, leucine, or valine [33]. In CsV-KB1, this position is replaced with a cysteine. The second conserved threonine (position 4 of the motif) is occasionally a serine or cysteine [33], but in CsV-KB1, it is substituted with a leucine, which is unusual for K^+^ channels. A BLAST search shows that a leucine is not present at this position in the consensus sequence of human K^+^ channels. In all but one of the sequences (RhiV-SA1), the amino acid immediately following the GYG is aspartate, which is found in the same position in many K^+^ channels [7,33], with the exception of those in which the tyrosine of GYG is substituted with a phenylalanine (GFG).

In addition to having the consensus motif itself, all of the sequences have a cluster of aromatic amino acids upstream of the motif (Figure 1). This is consistent with structural requirements of a K^+^ channel pore in which aromatic side chains in this position keep the pore at the appropriate diameter for K^+^ transport [34]. Hence, aside from a few unusual residues in or immediately after the GYG motif in two of the sequences, it would appear at first glance that any of these sequences could form functional K^+^ channels. To better understand the relationship of these four putative K^+^ channels with similar proteins from other viruses, we constructed a multisequence alignment and a phylogenetic tree (Figure 2). The tree includes sequences from all known viral K^+^ channel prototypes ([8]; Appendix A). Because of the small size of the proteins, most nodes only have bootstrap values. However, the tree still suggests that sequences from phage proteins are separated from the proteins from eukaryote-infecting viruses. While TetV-1, CsV-KB1, and FloV-SA1 cluster with proteins from other mimivirids (AaV1, OLPV2, and YLPV2), RhiV-SA1 is closer to the chlorovirus channels.

We then searched among viral and non-viral sequences in the protein databank for candidates that are most similar to the putative channels from the four *Mimiviridae* isolates using BLAST (Table 1). The best hits within the viruses confirm the results of the phylogenetic tree, in that, the putative channels show a distinct relationship to channel proteins from freshwater and marine phycodnaviruses as well as to a putative channel from other members of the *Mimiviridae* (*Aureococcus anophagefferens* virus; AaV). The high degree of diversity among the mimivirid sequences already suggests multiple independent events of lateral gene transfer.

The BLAST results also show that the putative channel proteins share up to 40% sequence identity with bacterial proteins. Most interesting in this context is the similarity between the primary sequences (low E values, Table 1) of some of the viral channel candidates to proteins with unknown function in bacteria in the phylum Actinobacteria. These organisms are thought to be very ancient, dating back to a pre-oxygen atmosphere on earth ~2.7 billion years ago [36,37]. It is possible that the recurrent similarity between viral and Actinobacteria proteins and the fact that both Actinobacteria and NCLDVs have ancient ancestors [36,37,38,39] could provide clues on the evolutionary origin of the viral K^+^ channels.

In addition to the consensus motif, which is part of the selectivity filter, a functional K^+^ channel requires additional structural elements. This includes at least two transmembrane domains that are able to span the bilayer and a short α-helix upstream of the selectivity filter motif [33,34]. The latter forms the pore helix and is essential for the correct positioning of the filter motif in the ion conducting pore [34]. A visualization of the bacterial channel KcsA (Figure 3A) serves as a reference for the pore structure. KcsA was the first crystalized K^+^ channel and serves as a model structure for K^+^ channel pores with all the aforementioned essential structural elements [34]. The same building elements can be recognized in the primary sequence of viral K^+^ channels by structural prediction algorithms. In the case of the functional viral channel Kcv_PBCV1_ [7] these algorithms predict with high propensity two transmembrane domains and an α-helix upstream of the selectivity filter (Figure 3B). To examine whether the new sequences fulfill these requirements, we applied the same structural prediction algorithms. The results show that TetV-1 has the predicted transmembrane domains (Figure 3B) and required structural elements in the exact positions that are expected for a functional K^+^ channel (Figure 3C). The combination of the overall structural architecture of a K^+^ channel pore and the presence of a consensus sequence suggest that this protein could indeed be a functional K^+^ channel.

The same analysis conducted with all four mimivirid sequences shows a more diverse picture (Figure 4A). Based on TMHMM and the consensus results from four additional prediction programs, both TetV-1 and RhiV-SA1 are robustly predicted to have both transmembrane domain (TMD)1 and TMD2 in the expected positions. In FloV-SA1, all programs predict an upstream TMD1-like domain, but there is no consensus prediction of a TMD2. CsV-KB1 also has a predicted upstream TMD, but the predicted second TMD is in the wrong position with the selectivity filter in the center of the transmembrane helix. We then examined whether a structural prediction algorithm provides evidence for an α-helix region prior to the canonical filter motive of K^+^ channels (Figure 1 and Figure 4B). In a previous study on other viral K^+^ channels, the prediction of α-helices in this position was very robust [17]. The data show α-helices in the correct position in 3 of the 4 viral protein sequences; only CsV-KB1 lacked this structure.

Collectively, the data indicate that at least three of the four protein sequences have the major structural elements consistent with a K^+^ channel function. TetV-1 and RhiV-SA1 show the most convincing evidence. FloV-SA1 is somewhat less convincing, because of disagreements in the prediction of TMD2, but it has the other expected features. The CsV-KB1 protein seems unlikely to be a functional K^+^ channel; it shows some deviation from the consensus sequence of K^+^ channels and the bioinformatic scrutiny of the amino acid sequence reveals no strong evidence for some essential structural elements.

### 3.2. The TetV-1 Protein Has K^+^ Channel Function

To test all four proteins for channel function, they were synthesized in vitro into nanodiscs and, after purification, reconstituted into planar lipid bilayers. Typical channel fluctuations were routinely obtained with the 79 amino acid TetV-1 protein (Figure 5A). None of the other proteins generated any perceivable channel activity in multiple repetitions (Table 2). These results establish that TetV-1 is a protein with typical K^+^ channel functions and that the other proteins do not work under conditions in which most viral K^+^ channels function [6,12,22,40]. Since it is known that the activity of some ion channels depends on factors such as the presence of anionic lipids [41], we cannot completely exclude the possibility that these other three proteins might function as K^+^ channels under different conditions.

We then conducted a basic characterization of the TetV-1 K^+^ channel protein, named Ktv1 for K^+^ channel from Tetraselmis virus 1. The channel generated in a solution with 100 mM K^+^ on both sides of the membrane an asymmetric current/voltage (I/V) in which the conductance at positive voltages (57 ± 7 pS) was about 2 times larger than that a negative voltages (27 ± 5 pS) (Figure 5B). This value of unitary conductance is similar to that of other viral K^+^ channels [12]. The Ktv1 channel also exhibited an appreciable voltage dependency; it was nearly always open at negative voltages and exhibited increasingly long closed times at positive voltages (Figure 5A). The open probability/voltage (P_0_/V) plot shows that this results in a progressive decrease in channel activity with increasing positive voltages. If we assume that, like the other viral K^+^ channels, the Ktv1 channel inserts preferentially with the n- and c-terminus into the membrane [22,42], the low open probability at positive voltages would imply that the channel is an inward rectifier.

To test the selectivity of Ktv1, the K^+^ on the *cis* side of the bilayer was replaced by Na^+^ (Figure 6A). As a result, the channel activity was only visible at negative voltages. This means that the channel conducts K^+^ inward, but there is no Na^+^ outward current. The assumption that the channel has a high preference for K^+^ over Na^+^ was confirmed by experiments in which the K^+^ on the *trans* side was replaced by Na^+^. In this situation, only outward K^+^ current but no Na^+^ inward current was visible. The mean I/V relations obtained from experiments with K^+^/Na^+^ on different sides of the membrane (Figure 6B) confirm that the channel transports exclusively K^+^. The I/V curves do not intersect with the voltage axis indicating perfect selectivity of the channel for K^+^ over Na^+^. Hence in agreement with the structure of its selectivity filter, Ktv1 is a highly selective K^+^ channel.

K^+^ channels are typically blocked by Ba^2+^ in a voltage dependent manner [43,44]. To test whether Ktv1 also exhibited this Ba^2+^ sensitivity, channel activity was recorded in 100 mM KCl and BaCl_2_ added at 5 mM first to the *trans* and then also to the *cis* chamber. The results indicate that the presence of the divalent cation on the *trans* side had no effect on the open channel amplitude (Figure 6C) but greatly reduced the open probability of the channel. The effect of the blocker is seen in Figure 6C: In the absence of the blocker the channel typically exhibits only short closures, while in its presence only short openings are seen. This mode of block and the voltage dependency of this effect, which increased with negative voltages (Figure 6D), is typical for the Ba^2+^ block of K^+^ channels [44]. Further addition of Ba^2+^ to the *cis* chamber resulted in a complete block of channel activity at positive voltages (Figure 6D). This effect is also typical for K^+^ channels in which Ba^2+^ blocks the channel with a higher affinity from the cytosolic than from the external side of the protein [45].

In summary, the combination of structural and functional data confirms that the Ktv1 79 amino acid protein from virus TetV-1, a member of the family *Mimiviridae*, functions as an ion channel with the typical hallmarks of a K^+^ channel. Modeling indicates that Ktv1 has the same architecture as the KcsA channel with all the structural elements of a K^+^ channel pore. However, like all other functional viral K^+^ channels, the transmembrane helixes are significantly shorter than those of the reference channel. This is most obvious for the inner transmembrane domains, which form the so-called bundle crossing gate in KcsA [34]. In the case of Ktv1, they are too short to form this gate (Figure 3C).

## 4. Conclusions

We provide the first direct evidence of a functional K^+^ channel encoded by a member of *Nucleocytoviricota* outside of the family *Phycodnaviridae*. All four members of the *Mimiviridae* family that were examined here code for proteins in which the structural hallmarks of functional K^+^ channels were either completely or partially conserved. Hence, like other giant viruses, the two families not only share a set of core genes that are typical for *Nucleocytoviricota* but also share subsets of other genes [46]. Scrutiny of the proteins of interest revealed high conservation of the selectivity filter with the typical K^+^ channel consensus sequence plus the upstream aromatic amino acids and the common aspartic acid after the GYG motif. However, this central domain was the only region common to all four proteins. Such structural diversity among the proteins from *Mimiviridae* and their distinct similarities to K^+^ channels from *Phycodnaviridae* tentatively suggests independent events of lateral gene transfer. A long evolutionary history of these proteins is possible since these viruses and their hosts are ancient. However, these evolutionary details cannot be resolved given the small datasets available so far.

We demonstrated that channel activity having the typical functional features of highly evolved K^+^ channels from mammals was present in the virus-encoded channel named Ktv1. Ktv1 had a remarkable high selectivity for K^+^ over Na^+^, channel gating, and a distinct voltage dependent sensitivity to Ba^2+^ block. These results indicate that the protein is not generating an unspecific leak conductance in the membrane but rather has all the structural and functional features of a bona fide selective K^+^ channel. That the other three putative channels did not generate detectable K^+^ channel activity does not rule out a channel function of these proteins. From an experimental point of view, there are several reasons for why these proteins were not generating measurable channel activity in our recording system. Based on experience with other channels, it is possible, for example, that these proteins require a distinct composition of the lipid bilayer [41] or the correct thickness of the bilayer in relation to the length of the transmembrane domains [47] for function. It is also possible that they are not properly folded and/or inserted into the membrane in the in vitro translation system with the result that they are not functional in the host bilayer. Another reasonable explanation for the negative results is that the channels may have a very small unitary conductance and/or very high flicker type open probability, which is not resolved with the standard measuring equipment [48,49].

We find it intriguing that all viral genes encoding K^+^ channels in *Nucleocytoviricota* are present in viruses that infect algae even though they are not necessarily obligatory for these viruses [50]. The hosts for the viruses with K^+^ channels vary from unicellular to multicellular [18] and from freshwater to marine algae [17,20,23]. It is possible that the K^+^ channel provides a function in these viruses that is of particular benefit when infecting algal hosts. However, the apparent exclusivity may simply reflect the very limited phylogenetic diversity of non-algal protists that have been used to isolate viruses in the phylum *Nucleocytoviricota* thus far.

For the PBCV1/*Chlorella* system, it is well established that the activity of the viral K^+^ channel is crucial early during infection when the viral membrane fuses with the plasma membrane of the host [51]. This triggers a depolarization and a loss of osmolytes and water from the host, which in turn lowers the high internal turgor pressure of host and promotes ejection of the viral DNA into the host [51,52]. Since marine algae do not have the same high internal pressure of freshwater algae [53], it is unlikely that the same mechanism is employed by all viruses that code for K^+^ channels. This implies these proteins may display alternative functions in the infection/replication cycle. An attractive hypothesis is that the activity of viral channel could indirectly alter crucial cellular parameters such as pH or Ca^2+^ in order to favor the activity of viral proteins [54]. This could be a successful and parsimonious way for individual virus to take command over their large hosts in the early steps of infection. If generally useful, K^+^ channels may yet turn up in nucleocytoplasmic viruses infecting non-photosynthetic protists with additional targeted isolation and screening.

## Figures and Tables

**Figure 1 viruses-12-01107-f001:**
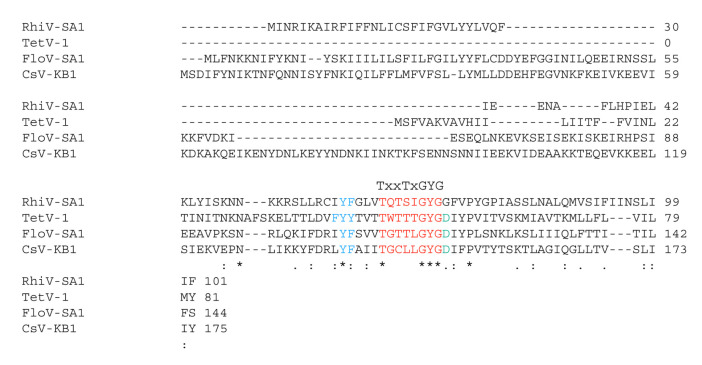
Multiple sequence alignment of putative K^+^ channels from viruses RhiV-SA1, TetV-1, FloV-SA-1, and CsV-KB1. The consensus sequence of K^+^ channels (on top) is highlighted in red. The residues in blue indicate functionally important aromatic amino acids in K^+^ channels upstream of the consensus motif. In the alignment, identical amino acids are indicated by “*” and conserved (amino acids with similar characteristics) and semi-conserved (amino acids having similar shape) amino acids are indicated by “:” and “.”, respectively.

**Figure 2 viruses-12-01107-f002:**
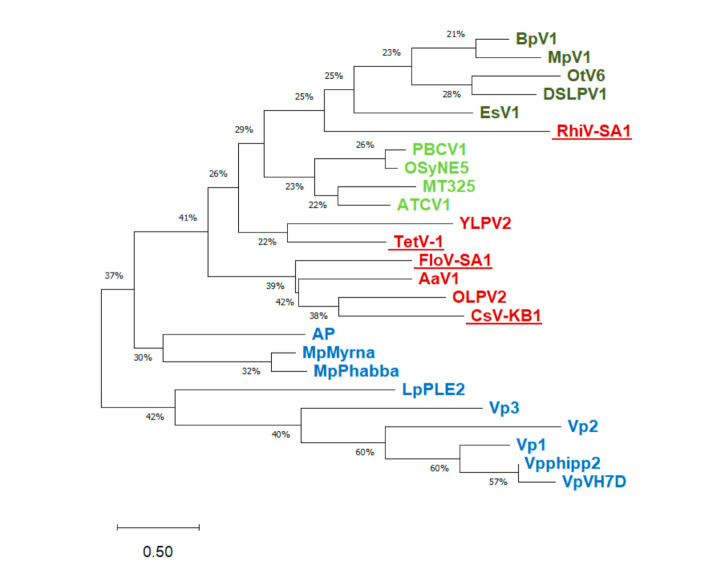
Phylogenetic tree of viral K^+^ channels including new sequences of putative K^+^ channels (underlined) from Figure 1. A possible evolutionary history was inferred by using the maximum-likelihood method and JTT matrix-based model [35]. The tree with the highest log likelihood (−7636.59) is shown. Initial trees for the heuristic search were obtained automatically by applying neighbor-join and BioNJ algorithms to a matrix of pairwise distances estimated using the JTT model, and then selecting the topology with superior log likelihood value. The tree is drawn to scale, with branch lengths measured in the number of substitutions per site. The proportion of sites where at least 1 unambiguous base is present in at least 1 sequence for each descendent clade is shown next to each internal node in the tree. This analysis involved 25 amino acid sequences. The final dataset contained a total of 366 positions. Evolutionary analyses were conducted in MEGA X [29]. Percentages are data coverage for each node. Color coding of the proteins is as follows: green = phycodnaviruses/chloroviruses, dark green = other phycodnaviruses, blue = phages, red = mimivirids.

**Figure 3 viruses-12-01107-f003:**
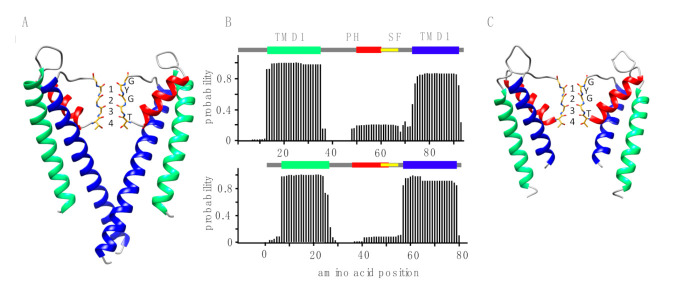
Structural elements of K^+^ channels. (**A**) Two of the four monomers of the crystal structure of KcsA channel (PDB ID: 1K4C) with outer (TMD1, green) and inner (TMD2, blue) transmembrane domains, pore helix (PH, red), and consensus sequence of the selectivity filter (SF, yellow). Atoms of the main chain of SF are shown. Atoms of the side chain of T75 are shown as it contributes to the K^+^ permeation pathway. Residues of the TXGYG consensus sequence are labeled, and the four K^+^ binding sites are numbered. (**B**) Structural predictions for the sequences of functional Kcv (K^+^ channel of chloroviruses) channel (Kcv_PBCV1_, top) and putative TetV-1 channel (bottom). The algorithm TMHMM predicts transmembrane domains with a high probability in both sequences (green and blue bars). From their location, they are equivalent to TMD1 and TMD2 of KcsA. The JPred algorithm predicts an α-helix (red) upstream of the selectivity filter in both proteins. This is equivalent to the pore helix of KcsA. The colors in B correspond to the colored structural elements in A. (**C**) Two of the four monomers of the model structure of the TetV-1 protein (based on the KcsA structure, PDB ID: 1K4C (https://www.rcsb.org/structure/1K4C)) with outer (TMD1, green) and inner (TMD2, blue) transmembrane domains, pore helix (PH, red), and consensus sequence of the selectivity filter (SF, yellow). Atoms of the main chain of SF are shown. Atoms of the side chain of T51 are shown as it contributes to the K^+^ permeation pathway. Residues of the TXGYG consensus sequence are labeled, and the four K^+^ binding sites are numbered.

**Figure 4 viruses-12-01107-f004:**
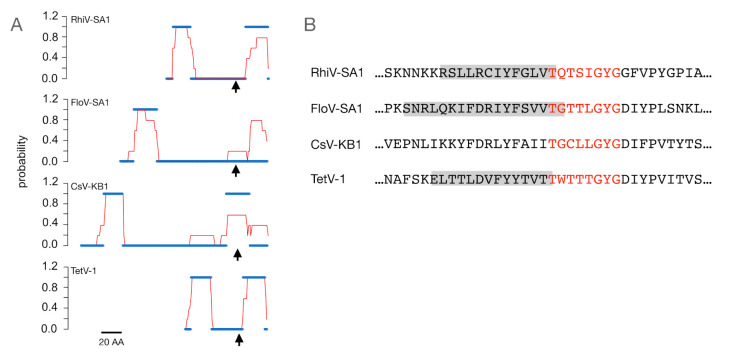
Structural predictions for putative K^+^ channel proteins. (**A**) Probability for transmembrane domains in four putative K^+^ channel sequences. The amino acid sequences from Figure 1 were analyzed with five different algorithms for predicting transmembrane domains. The red line shows the prediction by TMHMM and the blue line represents the mean of all predictions. For calculating the mean value, each amino acid which was predicted to be part of a transmembrane domain was set to 1 and all others to 0. The results are aligned to the position of the first glycine in the first G in the GYG motif (arrow). (**B**) Prediction of α-helices with the JPred algorithm. The grey bars indicate the predicted location of an α-helix upstream of the selectivity filter sequence (red) in 3 of the 4 sequences.

**Figure 5 viruses-12-01107-f005:**
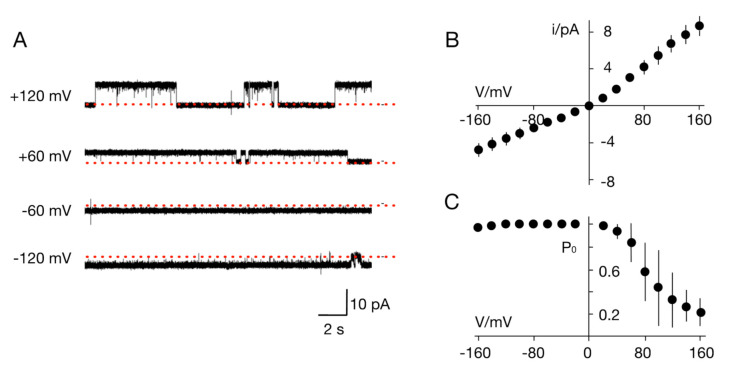
Ktv1 (K^+^ channel from Tetraselmis virus 1) is a functional K^+^ channel. (**A**) Reconstitution of the in vitro-synthesized proteins in planar phosphatidylcholine bilayers generates distinct fluctuations between closed (red line) and open states. The representative recordings in a symmetrical bilayer with 100 mM KCl at voltages between ±120 mV show frequent opening events at positive voltages and only rare closing events at negative voltages. Open channel current/voltage relation (**B**) and open probability/voltage (P_0_/V) plots (**C**). Data are the mean (± S.D.) of 3 independent recordings in same conditions as in A.

**Figure 6 viruses-12-01107-f006:**
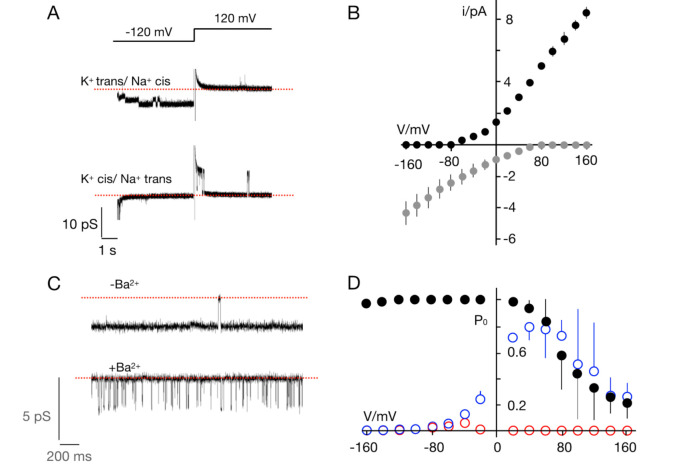
Ktv1 is K^+^ selective and blocked by Ba^2+^. (**A**) Current recordings of Ktv1 at −120 and +120 mV. Channel activity was measured with 100 mM KCl in *trans* and 100 mM NaCl in *cis* chamber (top) and in the inverse orientation (bottom). The dashed red line shows zero current level. Open channel currents are only recorded in the direction of the K^+^ fluxes. (**B**) Mean unitary channel current/voltage (I/V) relations for recordings with K^+^/Na^+^ on *cis/trans* (black symbols) or on *trans/cis* (grey symbols). (**C**) Channel fluctuations at −80 mV in symmetrical 100 mM KCl in the absence (−Ba^2+^) and presence (+Ba^2+^) of 5 mM BaCl_2_ in the *trans* chamber. The dashed red lines mark the closed state of the channel. (**D**) Probability/voltage (P_0_/V) plot from control measurements in symmetrical 100 mM KCl (black symbols, same as 5C) and after adding 5 mM BaCl_2_ to *trans* chamber (blue symbols) and to both chambers (red symbols). Data in B and C are means (± S.D.) of 3 independent recordings.

**Table 1 viruses-12-01107-t001:** Best non-virus and virus hits from BLAST searches of the four putative K^+^ channel candidates from viruses in the family *Mimiviridae* (aa = amino acid).

Source of Putative Channel Protein (NCBI Accession #)	Most Similar Non-Virus Hit, Organism Max Score, Total Score, Query Cover, E-Value, aa Identity, Accession #	Most Similar Virus Hit, Virus Max Score, Total Score, Query Cover, E-Value, aa Identity, Accession #
RhiV-SA1 (MT926120)	*Oleiphilus messinensis* (Proteobacteria) 49.7, 49.7, 72%, 2 × 10^−4^, 36%, WP_087462234.1	Acanthocystis turfacea chlorella virus mid_1.1 ATCV-1 (*Chlorovirus*) 43.9, 43.9, 46%, 4 × 10^−5^, 34%, QLC35876.1
FloV-SA1 (MT926122)	Actinomycetales bacterium (Actinobacteria) 77.8, 77.8, 88%, 3 × 10^−15^, 38%, PQM59864.1	Aureococcus anophagefferens virus (Aav) 58.2, 58.2, 90%, 1 × 10^−9^, 33%, YP_009052228.1
CsV-KB1 (MT926121)	Actinomycetales bacterium (Actinobacteria) 93.2, 93.2, 86%, 8 × 10^−21^, 36%, PQM59864.1	Organic Lake phycodnavirus 2 (OLPV2) 82.8, 82.8, 76%, 4 × 10^−19^, 39%, ADX06223.1
TetV-1 (AUF82121.1)	*Streptomyces laurentii* (Actinobacteria) 55.5, 55.5, 67%, 4 × 10^−7^, 42%, BAU83004.1	Ostreococcus tauri virus RT-201 40.8, 40.8, 61%, 4 × 10^−4^, 41%, AFC34969.1

**Table 2 viruses-12-01107-t002:** Ratio of measurements with channel activity (n_c_) divided by the number of attempts of reconstituting (n_a_) a putative channel protein in a bilayer of either DPhPC or DPhPG.

Source of Putative Channel Protein	n_c_/n_a_ in DPhPC	n_c_/n_a_ in DPhPG
RhiV-SA1	0/3	not tested
FloV-SA1	0/3	0/1
CsV-KB1	0/3	not tested
TetV-1	39/39	not tested

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
