# Peer review of "A Functional K+ Channel from Tetraselmis Virus 1, a Member of the Mimiviridae"

_viruses, 2020, doi:10.3390/v12101107_

Round 1

Reviewer 1 Report

The manuscript by Kukovetz et al. reported some genes related to K+ channels from Mimiviridae, a major lineage of large DNA viruses. Phylogenetic analysis based on this gene reproduced a separation of eukaryotic viruses from phages, and the eukaryotic viruses are also clustered into Mimiviridae and Phycodnaviridae expect for a Mimiviruse, Rhizochromulina sp. virus SA1. They demonstrated the channel activity of a K+ channel protein from the TetV using in-vitro protein synthesis and voltage clamp recording.    This paper is interesting, well written and the story is clear.  However, I have a series of recommendations that I hope may further improve the manuscript.     Title: Please consider to include the species name of the virus (ie. Tetraselmis virus)   L24 and elsewhere: “Mimiviridae” would be better as “mimivirid” is not commonly used.     L90: Please describe the program (software) used to predict the putative genes and how to distinguish them as K+ channel protein gene (It looks that MT926120–MT926122 in Table 1 are not derived from the database).   L146: Please define “conserved” and “semi-conserved” in this context.   L190: Please add a link (or reference) of the PDB.   L191: BLASTP?   L194: Since the phylogenetic tree doesn’t include cellular organisms, It is not clear that K+ channel genes were derived from a single origin or obtained by lateral gene transfer from their hosts.    L216: It is not clear what is the “similarities” stands for (similarity of what?). Please specify this.   L351: It is related to the comment above, I’m not completely sure that K+ channel gene is derived from a single origin (the common ancestor) of Phycodnaviridae and Mimiviridae, considering the small number of lineages used in the tree (Fig. 2) and the sequences are more similar to the cellular organisms than viruses (Table 1). It may be further verified by comparing the topology of the phylogenetic tree based on more conserved genes such as DNA polB and MCP. If  K+ channel genes are dispersed randomly across NCLDV phylogeny, it is more likely from the lateral gene transfer.   L363: Do the authors have any idea about the potential merit of K+ channel as an auxiliary metabolic gene on algae?    Table 2: It is not clear if channel activities of DPhPG were tested against the other 3 proteins as these cells are blank. Better to write “not tested” if needed.   Supplementary Table 1: I recommend this material to be included in the main manuscript.

Author Response

1st reviewer

1)Title: Please consider to include the species name of the virus (ie. Tetraselmis virus)  

Our Responds: we agree with this suggestion and we have changed the title accordingly.

2) L24 and elsewhere: “Mimiviridae” would be better as “mimivirid” is not commonly used.  

Our responds: also in this point we agree and we have exchanged mimivirid for Mimiviridae throughout the manuscript.

3) L90: Please describe the program (software) used to predict the putative genes and how to distinguish them as K+ channel protein gene (It looks that MT926120–MT926122 in Table 1 are not derived from the database).  

Our responds: We have expanded the materials and methods section with a list of all the algorithms, which were used. We also mention the source of the data.

4) L146: Please define “conserved” and “semi-conserved” in this context.  

Our responds: We have added a definition in the figure legend.

5) L190: Please add a link (or reference) of the PDB.

Our responds: we have added the link for the PDB in the figure legend

6) L191: BLASTP?  

Our responds: we have corrected BLAST into BLASTP and added our analytical procedure in Materials and Methods including a link.

7) L194: Since the phylogenetic tree doesn’t include cellular organisms, It is not clear that K+ channel genes were derived from a single origin or obtained by lateral gene transfer from their hosts.   

Our responds: we agree with the reviewer and we mention that the diversity among the viral channels is more of an indication for several incidences of lateral gene transfer. We also indicate that the low number of proteins in data banks is not yet sufficient for a definite answer to the question on the channel origin.

8) L216: It is not clear what is the “similarities” stands for (similarity of what?). Please specify this.  

Our responds: We have changed the wording and we include the notion of the E value as a parameters that defines similarities between primary sequences.

 9) L351: It is related to the comment above, I’m not completely sure that K+ channel gene is derived from a single origin (the common ancestor) of Phycodnaviridae and Mimiviridae, considering the small number of lineages used in the tree (Fig. 2) and the sequences are more similar to the cellular organisms than viruses (Table 1). It may be further verified by comparing the topology of the phylogenetic tree based on more conserved genes such as DNA polB and MCP. If  K+ channel genes are dispersed randomly across NCLDV phylogeny, it is more likely from the lateral gene transfer.  

Our responds: see responds to comment 7.

10) L363: Do the authors have any idea about the potential merit of K+ channel as an auxiliary metabolic gene on algae?   

Our responds: We have added a paragraph at the end of the conclusion on the potential significance of the channel in the infection/replication cycle.

11) Table 2: It is not clear if channel activities of DPhPG were tested against the other 3 proteins as these cells are blank. Better to write “not tested” if needed.  

Our responds: we have added “not tested” as suggested.

12) Supplementary Table 1: I recommend this material to be included in the main manuscript.

Our responds: At this point we do not agree with the reviewer. The table includes only vocabulary, which is not essential for understanding the paper. The large size of the table in the main paper would not be proportional to its importance.

Reviewer 2 Report

Kukovetz et al present the study "A functional K+ channel from a mimivirid", an interesting evaluation of how widespread K+ channels are within the phylum Nucleocytoviricota. 

Kukovetz et al present the study "A functional K+ channel from a mimivirid", an interesting evaluation of how widespread K+ channels are within the phylum Nucleocytoviricota.   

The authors sought to answer the following question: Are K+ channel encoding genes present in the phylum Nucleocytoviricota beyond the family phycodnaviridae, eg in mimiviridae, following up on recent findings that some mimivirids (AaV, TetV1, OLPV2) contain putative K channel pumps, but gene annotation alone does not guarantee a gene product performs its predicted function. The authors performed sequence analysis/alignment to identify 4 putative K+ channels from marine phytoplankton viruses TetV-1, CsV-KB1, FloV-SA1, RiV-SA1, followed by cell free in vivo synthesis (Membrane Max protein expression kit) of these 4 putative K+ channels in the presence of nanodiscs fllowed by channel D complex purification by metal chelate affinity chromatography, then functional testing with ion channel recording using planar lipid bilayer experiments.

The authors found low sequence homology among mimivirid putative K+ channels, but sequences consistent with K+ channel selectivity filter, and phylogenetic separation between sequences of K+ channels from phages vs eukaryote-infecting viruses. Interestingly, the authors find similarity between the putative mimivirid K+ channels and Actinobacterial proteins of unknown function, which may ultimately provide a lead toward identifying the source of these K+ channels. This information could benefit not only mimivirid research but also understanding of actinobacteria, a phylum with many ecologically and also medically important bacteria.

Functional testing revealed typical channel fluctuations for the TetV-1 protein, but no perceivable channel activity for the other three putative mimivirid K+ channels. The authors hypothesize that the other 3 channels may function as K+ channels under different conditions. Basic characterization identified the TetV-1 K+ channel, named K+ channel from Tetraselmis virus 1, or Ktv1, as an inward rectifier, with Ba2+ block behavior typical of K+ channels.

The overall message of the manuscript is identification of genes encoding functional K+ channels beyond phycodnaviridae, into mimiviridae, indicating a possible common origin, and implying a possible general, yet uncharacterized, physiological need of algae.  

Comments to authors: 

The article is extremely well written, with the rationale, hypothesis, methods, results, and interpretation clearly presented. 

My suggestions are minor:

1) I would suggest expanding the rationale for study of these algae-infecting viruses: beyond general interest, aren't some of the algae a food source for aquatic life / diversity ? Why is it important for us to characterize the viruses infecting aquatic algae, beyond using them to understand evolution ?  

2) The authors embark on characterizing 4 putative K+ channels, then find that only one appears to function as expected. A brief explanation is provided (need anionic lipids), could the authors add 1-2 sentences expanding on this a bit more ?  

3) Please provide a bit more information / citation on why viruses need K+ channels (The authors have previously described this: K+ channel deletion "reduced formation of viral progeny and viral pathogenicity, thus reducing the competitiveness and fitness of the virus (Takeda et al., 2002; Gonzalez and Carrasco, 2003"). A sentence or two providing updated hypothesis on why viruses need these channels during their lifecycle would be helpful for the reader.  

Author Response

2nd reviewer

My suggestions are minor:

  • I would suggest expanding the rationale for study of these algae-infecting viruses: beyond general interest, aren't some of the algae a food source for aquatic life / diversity ? Why is it important for us to characterize the viruses infecting aquatic algae, beyond using them to understand evolution ?  

Our responds: we have expanded the text and mention the importance of these channels, which are with their variable primary sequence, their small size and robust function an ideal model system for understanding basic structure/function correlates in K+ channel pores. This is exactly the main focus of research work.

2) The authors embark on characterizing 4 putative K+ channels, then find that only one appears to function as expected. A brief explanation is provided (need anionic lipids), could the authors add 1-2 sentences expanding on this a bit more ?  

Our responds: we have also expanded this part of the text and mention the limitations of the bilayer technique, which might be responsible for not finding channel function in our assay.

3) Please provide a bit more information / citation on why viruses need K+ channels (The authors have previously described this: K+ channel deletion "reduced formation of viral progeny and viral pathogenicity, thus reducing the competitiveness and fitness of the virus (Takeda et al., 2002; Gonzalez and Carrasco, 2003"). A sentence or two providing updated hypothesis on why viruses need these channels during their lifecycle would be helpful for the reader.  

Our responds: We have added a paragraph at the end of the conclusion on the potential significance of the channel in the infection/replication cycle. (See responds to point 10 of reviewer 1)